# The Role of Sleep in the Transition from Acute to Chronic Musculoskeletal Pain in Youth—A Narrative Review

**DOI:** 10.3390/children8030241

**Published:** 2021-03-20

**Authors:** Alessandro Andreucci, Cornelius B. Groenewald, Michael Skovdal Rathleff, Tonya M. Palermo

**Affiliations:** 1Center for General Practice, Aalborg University, 9220 Aalborg Øst, Denmark; misr@hst.aau.dk; 2Department of Anesthesiology and Pain Medicine, University of Washington School of Medicine, Seattle, WA 98105, USA; Cornelius.groenewald@seattlechildrens.org (C.B.G.); tonya.palermo@seattlechildrens.org (T.M.P.); 3Department of Health Science and Technology, Faculty of Medicine, Aalborg University, 9000 Aalborg, Denmark; 4Center for Child Health, Behavior and Development, Seattle Children’s Research Institute, Seattle, WA 98122, USA

**Keywords:** sleep, acute pain, adolescents, chronic pain

## Abstract

Musculoskeletal pain is common in the general pediatric population and is a challenge to youth, their parents, and society. The majority of children experiencing musculoskeletal pain will recover; however, a small subgroup of youth develops chronic pain. There is limited understanding of the factors that affect the transition from acute to chronic pain in youth. This review introduces sleep deficiency in the acute to chronic pain transition, exploring the potential mediational or mechanistic role and pathways of sleep in this process, including the interaction with sensory, psychological, and social components of pain and highlighting new avenues for treatment. Biological mechanisms include the increased production of inflammatory mediators and the effect on the hypothalamus-pituitary-adrenal (HPA) axis and on the dopaminergic signaling. Psychological and social components include the effect of sleep on the emotional-affective and behavioral components of pain, the negative impact on daily and social activities and coping strategies and on the reward system, increased pain catastrophizing, fear of pain, pain-related anxiety, hypervigilance, and social isolation. Future longitudinal studies are needed to elucidate these mechanistic pathways of the effect of sleep on the transition from acute to chronic pain, which may lead to the development of new treatment targets to prevent this transition.

## 1. Introduction

Musculoskeletal pain is common in the general pediatric population, with prevalence estimates of up to 40% in children and adolescents [1]. Most children with acute musculoskeletal pain recover [2]. However, up to 40% of adolescents develop musculoskeletal pain still present 5 years after onset [1,3]. Furthermore, rates of chronic pain are increasing, which places an increasing burden on both individuals and society [4,5]. The global impact of musculoskeletal pain in youth is underlined by back and neck pain, which are among the top 10 leading causes of years lived with disability (YLDs) in the world for adolescents [6]. This burden may be reduced if we successfully identify patient-related factors driving the transition from acute to chronic pain [2,7]. This knowledge could allow for the design and implementation of strategies aimed at preventing chronic pain.

While research on modifiable factors that might influence the transition from acute to chronic musculoskeletal pain [8,9] is rapidly evolving, current conceptual models of the transition from acute to chronic pain miss a potentially important modifiable factor in children and adolescents: sleep deficiency [10]. Sleep deficiency refers to problems in sleep quality or quantity and is a known predictor of acute pain following injury [11]. Sleep deficiency is common in children and adolescents, with prevalence estimates between 25% and 40% [12,13]. Moreover, sleep patterns have changed over the past few decades, and children now sleep 1 h less per day in most Western countries compared to 100 years ago [14]. Sleep has far ranging implications for children’s health and well-being.

Adolescence is a particular stage of life where a remodeling in sleep architecture occurs, with a reduction in slow wave sleep and rapid eye movement (REM) sleep. A shift in the biological mechanisms which modulate the circadian rhythms and sleep pressure occurs in adolescence, which coincides with a change in social activities, both resulting in later bedtimes [15]. Sleep deficiency has consistently been associated with chronic musculoskeletal pain in children and adolescents [16,17,18]. Indeed, 50–60% of adolescents with chronic pain problems report sleep deficiency [17]. The relationship between sleep deficiency and chronic pain has often been described as bi-directional, although evidence suggests that sleep is a stronger predictor of pain rather than vice versa [19,20,21]. However, despite this knowledge, sleep disturbance is not included in existing conceptual models of acute to chronic pain transition. Therefore, the aims in this narrative review are to (1) explore the potential mediational or mechanistic role and pathways of sleep in the acute to chronic pain transition, (2) briefly describe the current evidence regarding treatments for sleep problems in youth, and (3) provide an agenda for future targeted research in this area. 

## 2. Methods

A literature search was performed for the present narrative review. Original research and review articles related to the topic of this review were searched in the major electronic scientific databases (e.g., PubMed, Google Scholar) and summarized. Search terms entered in the scientific databases included: “sleep”, “acute pain”; “adolescents”; “chronic pain”, “musculoskeletal pain”; “mechanisms”; “mediation”; “transition”; “sleep treatment”; “psychological”; “affect”; “social”. Articles for which the full text was not available and were not in English or any other language for which the authors could provide translation (i.e., Italian, Spanish or Scandinavian languages) were excluded. The articles retrieved were complemented with articles already in possession of the authors of this review.

## 3. The Transition from Acute to Chronic Pain

While a full description of pain pathways is beyond the scope of this review, it is important to consider that noxious signals associated with acute musculoskeletal pain are carried via Aδ-and C-fibers to the dorsal horn of the spinal cord and then via ascending spinal neurons to the central nervous system, including the central sensory cortex [22,23]. In turn, the sensation of pain is modulated by a parallel descending pain inhibitory system. Maladaptive processes in any part of this pain pathway may drive acute to chronic pain transition [22,23,24] as described by Elman and Borsook [2,10], who constructed a model that describes the transition from acute to chronic pain by drawing partially from addiction neurobiology research. This model explains how contributory processes in the peripheral and central nervous system, including sensory, psychological (e.g., cognitions, emotions and motivations), and social components (e.g., interpersonal interactions), play a role in the evolution from tissue or nerve injury (initiating event), as detected by nociceptors (acute pain), in the transition to chronic pain [2,10]. According to this model, changes in the spinal cord and central nervous system would occur, which can result in hyperalgesia, allodynia, and a negative affect, as well as changes in several brain circuitries involved in this transition [10]. A key brain circuitry involved is the reward system. A feature of the reward system is the change in dopamine levels, which is also observed in pain conditions, although with differences between acute and chronic pain. Acute pain results in the activation of the reward system, which leads to a contingent increase in the levels of dopamine in brain regions such as the nucleus accumbens and the medial prefrontal cortex. This change in dopamine levels is modulated by mood, emotions, memory, stress, and also by the amygdala’s activity, which is responsible for providing an hedonic value and creating a memory of the pain event [2,10]. Other elements of reward also regulate the dopamine release, such as the opioid neurotransmission system and the homeostatic system, which compares the dopamine level to a threshold and serves as a biofeedback, thus modulating the dopamine production [10]. However, after prolonged dopamine production, the homeostatic system might become dysfunctional, resulting in neuroadaptations of the reward circuitry and impaired sensory, psychological, behavioral, and social functions leading to the development of chronic pain [10]. A number of other pathways are also described in this model. Despite its comprehensiveness, sleep was not included in this model, perhaps due to a lack of research attention at the time of its development. 

## 4. The Potential Role of Sleep in the Transition from Acute to Chronic Pain

Recent advances give a plausible explanation for how the inclusion of sleep may extend the model developed by Elman and Borsook. In the following paragraphs, we describe evidence highlighting how biological (Section 4.1), psychological (Section 4.2), and social (Section 4.3) components interact with sleep and pain to drive chronic pain development [19]. A summary of the effect of sleep deprivation on these components is outlined in Figure 1. Specific mechanisms that might explain the transition are further outlined within the following sections.

### 4.1. Sleep and the Sensory Component of Pain

#### 4.1.1. Evidence on the Effect of Sleep on the Sensory Component of Pain

Evidence (mainly from studies in adult populations) has shown that pain processing is altered in individuals who are sleep-deprived, with resulting lower pain thresholds and increased pain sensitivity [19,25]. A recent meta-analysis including 11 experimental studies of healthy adult individuals showed that sleep deprivation increases self-reported pain perception and influences pain responses to somatosensory tests (e.g., pressure pain thresholds, heat pain thresholds, cold pain thresholds) [25]. A more recent study (not included in that meta-analysis) showed impaired conditioned pain modulation, increased pain sensitivity to pressure and cold pain, and facilitated temporal summation of pain in young healthy adults after 24 h of total sleep deprivation [26]. Despite the body of evidence primarily originating from adult populations, new research indicates these findings are also applicable to the adolescent population. Recently, Krietsch showed in a three-week cross-over experimental study in which adolescents also reported greater frequency of pain complaints following 5 nights of insufficient sleep [27]. 

#### 4.1.2. Mechanisms to Explain the Effect of Sleep on the Sensory Component of Pain

There are several potential biological mechanisms contributing to the sleep-pain association. This might involve the increased production of inflammatory mediators such as prostaglandins and cytokines, observed in sleep-deprived individuals, as well as in different pain conditions [28]. Other substances such as serotonin, norepinephrine, and adenosine, as well as the orexinergic system, might be involved. These substances modulate pain but also regulate the sleep-wake cycle [15,28,29,30]. Findings from both human and animal studies suggests that sleep deprivation might lead to the alteration of the serotonergic system’s functioning [28,30]. This would impair the functioning of the serotoninergic descending inhibitory system and decrease the pain threshold [30]. However, it should be noted that there are differences between the effect of acute (increased serotonin production) and chronic (reduced sensitivity of serotonin 1A receptor) sleep deprivation, and further studies are needed [28]. Norepinephrine promotes wakefulness, and increased norepinephrine levels are observed in sleep-deprived individuals [28]. However, how norepinephrine can lead to pain in sleep-deprived individuals has not been clarified yet, as previous preclinical studies suggest that higher norepinephrine levels are associated with an analgesic effect [28]. Adenosine regulates sleep, and adenosine levels increase during prolonged wake time. Findings from studies in rats showed that increased adenosine levels were associated with both sleep deprivation and hyperalgesia, suggesting that increased adenosine levels might mediate the effect of sleep on pain [28]. The orexinergic system includes two neuropeptides (orexin A and B). Increased orexin activity is associated with the wake status; in addition, the orexinergic system is also involved in the modulation of pain. Therefore, it has been proposed as a mediator of the effect of sleep deprivation on pain, although further studies are needed to elucidate this [28]. 

While a detailed discussion of these systems is beyond the scope of this review, they share the common characteristics of being proinflammatory in response to acute musculoskeletal injury and, in turn, further activated in response to sleep deprivation. In addition, the functioning of the hypothalamus-pituitary-adrenal (HPA) axis is involved in the response to physical and psychological stress, and also in the regulation of sleep [15,28,31]. Acute musculoskeletal injury triggers the amygdala in the activation of the HPA axis, resulting in the production of cortisol, a stress-response hormone which is normally rapidly cleared from the body once stress resolves [15,28,31]. However, a maladaptive response to acute stress may cause a prolonged stimulation of the HPA axis, which in the long-term could become dysfunctional and hyper-reactive to stressful stimuli. This has been observed in chronic pain conditions and also in individuals with insomnia, who showed greater cortisol production and an over-reactivity of the HPA axis, which mediated the association between sleep deficiency and pain sensitivity [28,31,32,33]. Another potential mechanism involves the dopaminergic signaling system, which is central to pain transmission. Endogenous dopamine production is associated with analgesia in the presence of acute pain [34,35,36]. However, dopamine function is reduced by sleep disruption [28], which may contribute to the hypodopaminergic state found in some individuals with chronic pain [10,36]. Lastly, sleep deficiency also affects the opioidergic system, which could play a role in acute to chronic pain transition. Evidence suggests a bi-directional relationship between sleep deprivation and opioid use [29]. In individuals with acute pain following injury or surgery, sleep deprivation could predispose them to a maladaptive descending inhibitory control of pain involving the opioidergic system (i.e., downregulation of µ and δ opioid receptors and reduced basal levels of endogenous opioid levels in the central nervous system) [19,37,38,39,40,41,42,43,44], leading to increased opioid use for uncontrolled pain, while increased prescription opioid use could disrupt sleep architecture and increase daytime sleepiness and somnolence [29,45,46,47]. Furthermore, sleep disturbance is a hallmark of chronic opioid use, contributes to opioid withdrawal, and is an important reason for relapse in those recovering from opioid abuse [48,49]. The above-described mechanisms are summarized in Table 1.

### 4.2. Sleep and Psychological Factors

#### 4.2.1. Evidence on the Effect of Sleep on the Psychological Component of Pain

Sleep deficiency, psychological distress (e.g., depression, anxiety, stress) and chronic pain often co-occur [15,17,35]. Sleep deficiency can directly influence mood and affect, including increasing levels of depression, anxiety, and stress and by affecting cognitive and emotional pathways, as sleep itself regulates emotions and motivation [15,34,50,51,52]. Recent evidence suggests that the interaction between sleep and psychological factors can affect the transition from acute to chronic pain. Prior reviews of adult studies found that negative affect mediated the relationship between sleep and pain [19,33,36]. The evidence in children is less exhaustive. In a cross-sectional study with children with chronic pain the association between poor sleep quality and pain was partially mediated by negative affect (but not positive affect) [53]. In a longitudinal study in children with sickle cell disease the association between poor sleep quality and pain intensity was partially mediated by negative mood [54]. Another study (cross-sectional) including children with chronic pain showed that the association between sleep quality and pain outcomes (intensity, interference) was partially mediated by anxiety and depression [55]. Results from a longitudinal study including children with polyarticular arthritis suggest that decreased positive affect (e.g., cheerfulness, calmness) may mediate the relationship between sleep and chronic pain [56]. High levels of pain catastrophizing experienced in individuals with acute pain and sleep deficiency have been proposed as a mediator of the transition to chronic pain [15,36], but no studies in children have tested this hypothesis by means of mediational analysis. 

#### 4.2.2. Mechanisms to Explain the Effect of Sleep on the Psychological Component of Pain

There are multiple pathways by which the effect of sleep on psychological factors affects acute pain perception and the transition to chronic pain [15,36]. Potential mechanisms include an effect of negative mood on the emotional-affective and behavioral components of pain that would lead to a decrease in the pain threshold [30] and a negative impact on the engagement in daily activities and an increased use of maladaptive coping strategies [15]. Cognitive appraisals such as pain catastrophizing, where sleep deprivation is caused by excessive rumination and, in turn, worse sleep increases catastrophizing thoughts about pain, might be another important psychological factor influencing the transition to chronic pain in youth, as observed in adults with chronic pain [15,36].

In addition, sleep deprivation affects the reward system, with differences between acute (increased motivation for reward) and chronic sleep deprivation (which might lead to both an increase or a decrease in motivation for reward) [34,50]. A dysfunctional reward system might therefore predispose to the development of chronic pain. Neurotransmitters involved in mood/affect, sleep, and pain include the dopaminergic signaling pathway [35] and 5-hydroxytryptamine (5-HT) neuromodulators [30]. These neuromodulators are involved in the modulation of the nociceptive stimulus, and a decreased functional availability observed in depressive mood, sleep deficiency, and chronic pain status might be another pathway for the role of sleep and depression in the transition from acute to chronic pain [19,30,35]. The amygdala activation might be another pathway. The amygdala is involved in pain modulation and in creating an emotional link to pain when a painful event occurs. Sleep deprivation affects the amygdala activity, which becomes hyperactive during sleep deprivation but also in depressed individuals [15,30,34]. It has been proposed that both positive and negative emotional states (stress, anxiety, and depression) might have a role in modulating the perception of pain through the amygdala. Negative emotions would lead to increased pain through the activation of the amygdala, while positive emotions would inhibit the amygdala and decrease pain [15]. A summary of these mechanisms is outlined in Table 2.

### 4.3. Sleep and Social Factors

#### 4.3.1. Evidence on the Effect of Sleep on the Social Component of Pain

Sleep deprivation can affect daily life for youth in terms of social interactions, isolation, loneliness, and the ability to carry out normal daily activities [15,57]. For example, in observational studies in adolescents, sleep deficiency was associated with a decreased ability to carry out activities in the domains of school, recreation, and social activities [58,59]. Conversely, improved bedtime routines, better sleep quality, and later school times were associated with improved school attendance and social functioning in children and adolescents [60,61,62]. The presence of pain itself can also result in a reduced frequency of social activities and therefore in more difficulties in developing or maintaining strong peer relationships, a higher dependency on the support of the family, and limitations in performing daily activities and physical activity [63,64]. Individuals experiencing musculoskeletal pain might also experience more restrictions in attending school, further straining social relationships [63,64]. In turn, factors related to the social sphere (e.g., social isolation, social functioning, loneliness) are associated with increased pain perception, musculoskeletal pain frequency, and lower levels of physical function [65,66]. This might result in the creation of a vicious cycle between musculoskeletal pain and social isolation [64]. The limitations in daily activities and reduced social interactions experienced by individuals with pain can be further exacerbated by sleep deprivation, as shown in previous studies [67]. However, studies in children specifically assessing social factors as mediators of the association between sleep and chronic pain have not been conducted yet, and further studies are needed.

#### 4.3.2. Mechanisms to Explain the Effect of Sleep on the Social Component of Pain

There are different proposed mechanisms to explain how the effect of sleep deprivation on social components influences the transition from acute to chronic musculoskeletal pain. Sleep deprivation exacerbates the negative reactivity to emotional stimuli and increases social isolation in individuals with acute pain [57,58]. The amygdala activation is a potential mechanism to explain the transition to chronic pain in children with acute pain who experience social isolation and increased emotional reactivity. Studies have shown hyper-activation of the amygdala in response to emotional stimuli in sleep-deprived individuals [57]. Increased amygdala activity has also been associated with the loss of regulatory control and excessive emotional sensitivity [34]. The increased amygdala activity may thus increase the attention toward pain, through its effect on the reward system. In addition, the negative regulation of emotional stimuli and social isolation may further interfere with a youth’s ability to cope with pain and increase pain perception, thus potentially driving the transition from acute to chronic pain [15]. Sleep deprivation might further impair a youth’s ability to perform meaningful daily activities (including social, recreational, and physical activity) [3,59,68]. In the long-term, this might lead to greater disability and the development of chronic musculoskeletal pain [3]. In addition, co-occurring pain and sleep deprivation might result in persistent fatigue and lack of concentration [69]. This can lead to further disability in youth and contribute to the development of chronic pain. Additionally, sleep deprived youths who are more socially isolated might experience more difficulties in ignoring pain and might develop behaviors that lead to an increased focus on pain (i.e., fear of pain, pain-related anxiety, and hypervigilance, as proposed by the Fear-Avoidance model of pain) [33,69,70] and to the avoidance of social activities that are perceived as painful [59,67] and would affect coping strategies. Overall, this can contribute to the maintenance of pain and disability [10,69,70]. The above-described mechanisms are summarized in Table 3.

## 5. The Role of Sleep Treatments 

While not yet studied in the context of preventing acute to chronic pain transmission in youth, several sleep treatments could be helpful in this context. These include cognitive-behavioral interventions and pharmacotherapy, which should be tailored to the specific type of sleep problem and might be used alone or in combination [62,71,72]. A brief summary of these treatments is outlined below with a focus on the treatment of insomnia, which is the most common sleep complaint among youths with pain [16,72]. 

### 5.1. Cognitive-Behavioral Interventions

Cognitive-behavioral Therapy for Insomnia (CBT-I) is the gold standard treatment for adults experiencing insomnia and can be modified for older children and adolescents [16,72]. CBT-I is a multicomponent treatment, targeting a range of cognitive, behavioral, and social factors that may perpetuate insomnia, such as poor sleep hygiene practices (e.g., too much time in bed, watching TV in bed) and negative attitudes about sleep (e.g., concern about daytime deficits) [16,71,72]. The core behavioral skills are stimulus control and sleep restriction. Stimulus control involves keeping a regular sleep schedule with sleep limited to the night time, in addition to the improvement of the bed-sleep association (e.g., going to bed only when sleepy, using the bed only for sleep and not for other stimulating activities such as watching TV and social media) [62,71,72]. Sleep restriction involves creating a sleep window to increase the time spent asleep while in bed and minimizing the time spent awake in bed. This is accomplished by temporarily reducing the length of the sleep period (based on sleep diaries indicating sleep and wake times). The aim is to improve the association between bed and sleep (going to bed only when almost asleep) and increase sleep efficiency [62,71].

Relaxation techniques include meditation, deep breathing, and muscle relaxation and imagery, which consists in visualizing positive images such as relaxing scenes [71,72]. Strategies to improve sleep hygiene may focus on establishing a night-time routine and a regular sleep schedule with a consistent wake-up time, sleeping in a cool, dark and quiet environment, avoiding the consumption of food and drinks that contain caffeine, and avoiding screen time before sleep [13,62].

While there is robust evidence for both the short-term and long-term improvements in sleep following CBT-I in adults, there has been fewer studies of CBT-I conducted in adolescents [13,71]. A recent systematic review of randomized controlled trials (RCTs) of CBT-I among adolescents reported improvements in subjective (perceived sleep quality) and objective sleep parameters (total sleep time, sleep onset latency, and sleep efficiency). However, the RCTs had several limitations, such as a small sample size, a short follow-up, and the exclusion of adolescents with comorbid physical and mental health conditions [73]. Preliminary new evidence shows that CBT-I can be adapted for populations with chronic pain, with promising results [16]. In a pilot trial including adolescents with insomnia and comorbid physical or psychological difficulties (chronic pain, anxiety, and depression), CBT-I was associated with significant moderate to large improvements in self-reported measures of sleep, anxiety, depression, and quality of life, and the effects were maintained after 3 months [74]. In that trial, the CBT-I protocol included four sessions (max. 75 min each) delivered over 4 to 6 weeks, in addition to the completion of a sleep diary during the study period. The intervention strategies were: sleep education, sleep hygiene training, sleep restriction, and stimulus control [74]. In another pilot trial including adolescents with headache and insomnia, hybrid CBT-I (targeting both sleep and headache) improved sleep and reduced headache frequency at a 3-month follow-up [75]. The protocol comprised six sessions including core CBT-I strategies (stimulus, control, sleep restriction, sleep hygiene training) and headache education, relaxation training, pleasant activity scheduling and positive thought tracking, and parent operant training (to improve adolescent skills practice). The results of this trial are in line with other hybrid interventions tested in adults including both sleep and pain treatment components [76]. 

### 5.2. Pharmacotherapy

Pharmacotherapy (both over-the-counter and prescribed medication) is commonly used to treat pediatric sleep disturbances and may include sleep medications such as benzodiazepine receptor agonists (zolpidem, zaleplon, and eszopiclone), alpha agonists (clonidine, guanfacine), antihistamines, melatonin, and chloral hydrate [62,71,72]. However, it is not recommended as a first-line strategy for the treatment of sleep problems, due to the lack of evidence on the long-term efficacy and safety of pharmacotherapy for sleep in children and adolescents, despite the common use [72,77]. Indeed, there are concerns regarding potential risks of dependency, rebound effects, addiction, and withdrawal complications of these medications [77]. Therefore, recommendations are to limit pharmacotherapy and use only together with other behavioral approaches [62,71].

## 6. Limitations

This narrative review has several limitations. A systematic approach was not used for searching the evidence base, which would have enabled an unbiased selection of the literature. In addition, only a few scientific databases were searched. Therefore, it might be argued that some literature on potential mediators of sleep in the transition from acute to chronic pain were not found and discussed. However, this is unlikely, as we included and discussed evidence from the most recent reviews on sleep and pain.

## 7. Research Agenda for Future Investigations

We propose that future research focus on understanding the role of sleep in the transition from acute to chronic musculoskeletal pain in children and adolescents. Several specific topics warrant further investigation:Longitudinal or experimental studies are needed to understand the mediational role of modifiable social and psychological factors on the effect of sleep in the transition from acute to chronic musculoskeletal pain. The evidence for social and psychological factors in children and adolescents is based on only a few cross-sectional studies [53,55] and longitudinal studies [54,56]. Additional longitudinal studies are needed to evaluate the temporal and causal relationships between sleep, social and psychological factors, and subsequent acute to chronic pain transition. This knowledge would allow for a better understanding of the temporal sequence of mechanisms underlying the sleep-pain association, and identify mediators that may be targeted by interventions in order to prevent the development of chronic pain (e.g., reducing social isolation). Investigations of youth following musculoskeletal injury may provide the ideal means to prospectively study these factors over the course of the evolving pain experience.Further research is needed to clarify the role of biological factors by means of experimental studies that assess the change in inflammatory mediators (e.g., prostaglandins, cytokines), other substances and neurotransmitters (serotonin, norepinephrine, adenosine, orexin), and HPA axis functioning and their association with pain in children with sleep deficiency. This would allow for a better understanding of the role of biological mechanisms in the transition to chronic pain and their specific intersection with sleep.There is also an urgent need for research evaluating whether existing treatments aimed at improving sleep in youth may extend to the context of preventing acute to chronic pain transition. For example, CBT-I has shown efficacy for sleep improvement in multiple pain populations [16], but its role in preventing acute to chronic pain transition remains unexplored.

## 8. Conclusions

This review has described the role of sleep in the transition from acute to chronic pain in youth. In light of the profound negative effects of sleep deficiency on youth health and functioning, future research efforts should focus on sleep as a potentially important target for intervention to prevent the transition from acute to chronic pain. Potential mechanistic pathways of the effect of sleep on this transition have been outlined and cover biological, psychological, and social factors that could be targeted to prevent the transition from acute to chronic pain. Future longitudinal studies with multiple time-points are needed to elucidate the mechanisms through which sleep affects the transition from acute to chronic pain. The evidence regarding treatment options for sleep is presented and future research should focus on therapies targeting both sleep and pain in adolescent populations to prevent the transition from acute to chronic pain.

## Figures and Tables

**Figure 1 children-08-00241-f001:**
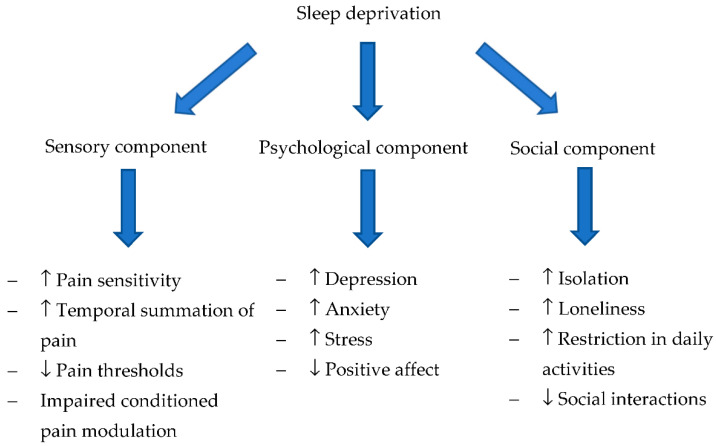
Effects of sleep deprivation on sensory, psychological, and social components involved in the transition to chronic pain. ↑ = Increased; ↓ = Decreased.

**Table 1 children-08-00241-t001:** Summary of mechanisms to explain the effect of sleep on the sensory component of pain.

Proposed Mechanism	Relationship with Sleep	Relationship with Pain
↑ Inflammatory mediators (prostaglandins, cytokines)	↑ Prostaglandins, cytokines after SD	↑ Prostaglandins, cytokines in pain conditions
Altered serotonergic system	↑ Serotonin (acute SD) ↓ Serotonin 1A receptor sensitivity (chronic SD)	Impaired serotonergic descending inhibitory system ↓ Pain threshold
↑ Norepinephrine	↑ Norepinephrine after SD	Not clear yet
↑ Adenosine	↑ Adenosine after SD	↑ Adenosine associated with hyperalgesia
↑ Activity orexinergic system	↑ Activity orexinergic system after SD	Pain modulation
HPA axis	↑ HPA axis reactivity after SD	↑ HPA axis reactivity in chronic pain
Dopaminergic signaling	↓ Dopamine after SD	↓ Dopamine in chronic pain
Opioidergic system	↓ Endogenous opioids after SD ↓ µ and δ opioid receptors	Impaired opioidergic descending inhibitory system

SD = Sleep deprivation; HPA axis = Hypothalamus-pituitary-adrenal axis; ↑ = Increased; ↓ = Decreased.

**Table 2 children-08-00241-t002:** Summary of mechanisms to explain the effect of sleep on the psychological component of pain.

Proposed Mechanism	Relationship with Sleep	Relationship with Pain
Emotional component of pain	Negative mood after SD affects the emotional component of pain	↓ Pain threshold due to high negative mood
Coping strategies	Maladaptive coping after SD	↑ Pain perception due to maladaptive coping strategies
Engagement in daily activities	↓ Engagement in daily activities after SD	↓ Engagement in daily activities linked to pain
Catastrophizing	↑ Catastrophizing after SD	↑ Focus on pain due to catastrophizing
Reward	↑ Motivation after acute SD ↑ or ↓ motivation after chronic SD	Dysfunctional reward associated with chronic pain
5-HT neuromodulators	↓ 5-HT both in depression and after SD	↓ 5-HT in chronic pain
Dopaminergic signaling	↓ Dopamine both in depression and after SD	↓ Dopamine in chronic pain
Amygdala	↑ Amygdala activity in negative emotional states and after SD ↓ Amygdala activity in positive emotional states	↑ Amygdala activity - increased pain ↓ Amygdala activity - decreased pain

SD = Sleep deprivation; ↑ = Increased; ↓ = Decreased.

**Table 3 children-08-00241-t003:** Summary of mechanisms to explain the effect of sleep on the social component of pain.

Proposed Mechanism	Relationship with Sleep	Relationship with Pain
Maladaptive coping	Maladaptive coping after SD in socially isolated children	↑ Pain perception
↓ Engagement in daily activities	↓ Engagement in daily activities after SD and social isolation	↑ Disability and chronic pain
Fatigue	↑ Fatigue after SD	↑ Disability and chronic pain
Fear-avoidant behaviors	↑ Fear-avoidant behaviors after SD	↑ Disability and chronic pain
Amygdala	↑ Amygdala activity after SD in socially isolated children	↑ Attention to pain due to: ↑ emotional sensitivity ↓ regulatory control

SD = Sleep deprivation; ↑ = Increased; ↓ = Decreased.

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
