# Peer review of "The Role of Sleep in the Transition from Acute to Chronic Musculoskeletal Pain in Youth—A Narrative Review"

_children, 2021, doi:10.3390/children8030241_

Round 1
Reviewer 1 Report
In this narrative review, the Authors intended to summarize and explain the role of sleep in the transition from acute to chronic musculoskeletal pain in youth. The paper aimed to present recent evidence for the role of sleep deprivation in the transition of acute to chronic pain, explore the possible pathways, summarize available treatment options for sleep disorders in adolescence, and propose future research needs.
The structure of the paper is appropriate and appropriate for a sort of a mini-review. The review of the evidence is unstructured and random. At least, the Authors should discuss the key references more extensively. The best way would be to present the preclinical and clinical studies in a table.
The references are up-to-date. It is not justified to have 88 references for such a short manuscript. Please, note that there are eleven references of the last author cited.
The Introduction is appropriate and clear.
Chapter 2. The transition from acute to chronic pain (lines 66-80) refers to the model by Elman and Borsook, but it does not bring information about the process of acute to chronic pain transition. The reader does not receive any explanation. I strongly suggest developing this paragraph.
Chapter 3. 3. The potential role of sleep in the transition from acute to chronic pain (lines 82 and further) is the most important part of the manuscript. The Authors proposed a graphical presentation of the effects of sleep deprivation on sensory, psychological, and social components involved in the transition to chronic pain. I find this figure misleading and requiring further development. There are different categories mixed in one picture: the pathophysiological mechanisms of pain, the social aftermath of sleep deprivation, and the psychological consequences. Amygdala modulation is not a psychological factor. It is rather a mechanism involved and affected by sleep deficiency.
The logical relation of particular factors is missing.
This approach made your further chapters perceived as erratic and hard to understand in total. The facts you collected are appropriate but need re-organizing in order to elucidate the logical consequence from one to another. Otherwise, the reader will still be missing the answer to how sleep deprivation makes acute pain change to chronic.
E.g., lines 158-161 "Recent evidence suggests that the interaction between sleep and psychological factors can affect the transition from acute to chronic pain. Negative affect (e.g., depression, anxiety) has been proposed as mediating the relationship between sleep and pain in previous reviews (22,25,43,62)." I strongly suggest reviewing the evidence here. This is a clue to the paper.
E.g., Lines 213-214 "Sleep deprivation exacerbates negative reactivity to emotional stimuli and increases social isolation in individuals with acute pain." - consistently, you should lead to the emotional pathway and amygdala modulation.
The Authors discuss the biological mechanisms that contribute to the sleep-pain association. I strongly suggest organizing this chapter and add a figure or a table summarizing these mechanisms. Please note that the descending inhibitory pathways are associated with serotonin and norepinephrine as neurotransmitters, which should be at least mentioned, as other neurotransmitters are listed.
Chapter 4 on sleep treatments is highly insufficient. The approach is correct but should be re-ordered: 1) behavioral modifications, 2) CBT intervention, 3) pharmacotherapy. Please, write more on CBT in terms of behavioral interventions (examples).
Chapter 5 on the research agenda for future investigations does not result from the previous parts of the manuscript and may be perceived as separate Authors' postulates.
Overall, it is important to summarize and clarify the role of sleep deprivation in the process of acute to chronic pain transition. This paper is a kind of short narrative review. The structure is well-shaped. It needs an evidence-based approach and significant re-editing, though.
Reviewer 2 Report
Dear authors and editor,
The manuscript titled "The role of sleep in the transition from acute to chronic musculoskeletal pain in youth". This is a narrative review that introduces sleep deficiency in the acute to chronic pain transition, exploring the potential mediational or mechanistic role and pathways of sleep in this process, including the interaction with sensory, psychological and social components of pain and highlighting new avenues for treatment
There are many minor and major issues I'd like the authors resolve.
Title:
1-The indication ‘literature or narrative review’ or ‘review of the literature’ is helpful in clarifying the research design. It is recommended to add
Abstract:
2-Change the keywords. Delete the words "transition". Not found in the MeSH (Medical Subject Headings).
3- I recommend including a limitations section. Narrative reviews have a number of design limitations. Also, I recommend a small section on methods to reduce biases. As an example, a literature search was performedfor the present study on the lines of searches foran non-systematic or narrative review, but including features of systematic review methodology. The electronic search included three data-bases, PubMed, EMBASE and Google Scholar, and used search terms: "sleep", "acute pain"; "adolescents"; "chronic pain", "musculoskeletal pain" ...
The inclusion criteria were: all types of articles, articles publishedin PubMed, and related only to humans. The exclusion criteria were: articles for which full text was not available, were not in English, or were grey literature.
Aims
4-The aims are too demanding for the design used. It is recommended that the aims be rewritten.
References:
5-I recommend that you review the rules of the publisher MDPI
Round 2
Reviewer 1 Report
The article is now concise and clear. It reads well. Congratulations.
Reviewer 2 Report
Dears authors;
I am satified about revised version, it is suitable to be accepted now. However, it is a pity that such an interesting topic is based only on the search of two databases. A lot of information that could be relevant is lost.
Don´t forget to change the keywords. Delete the words "transition". Not found in the MeSH (Medical Subject Headings)
Kind regards.